# Culture-Dependent and -Independent Wastewater Surveillance for Multiple Pathogenic Yeasts

**DOI:** 10.3390/jof11020086

**Published:** 2025-01-23

**Authors:** Tyla Baker, Phillip Armand Bester, Olihile Moses Sebolai, Jacobus Albertyn, Carolina Henritta Pohl

**Affiliations:** 1Department of Microbiology and Biochemistry, University of the Free State, P.O. Box 339, Bloemfontein 9301, South Africa; bakertyla01@gmail.com (T.B.); sebolaiom@ufs.ac.za (O.M.S.); albertynj@ufs.ac.za (J.A.); 2School of Pathology, University of the Free State, P.O. Box 339, Bloemfontein 9301, South Africa; besterpa@ufs.ac.za

**Keywords:** pathogenic yeast, wastewater surveillance, multiplex PCR

## Abstract

Wastewater surveillance is a promising tool to monitor potential outbreaks and determine the disease burden within a community. This system has been extensively used to monitor polio and COVID-19 infection levels, yet few attempts have been made to apply it to monitoring pathogenic yeast. This study aimed to investigate the application of wastewater surveillance for potentially pathogenic yeast in wastewater treatment plant influent. This was done by comparing culture-dependent data with culture-independent data and investigating the fluconazole concentration in wastewater. Additional studies on the growth of isolated strains were conducted. We found that a multiplex PCR system to detect multiple yeasts holds promise as a molecular detection tool for wastewater surveillance. Culture-dependent results indicated that *Candida* spp. specifically *C. krusei* and *C. glabrata*, were most prominent. Growth studies supported that these species grow well in this environment while the less frequently isolated yeasts grew poorly. The data from culture-dependent and independent techniques showed some correlation, with similar species being identified with both, further promoting the use of molecular tools for surveillance. This study highlights the presence of potentially pathogenic yeasts in wastewater, which may indicate the prevalence of these yeasts in the environment or community. This wastewater may also be a potential source of infection for persons encountering it due to poor wastewater management.

## 1. Introduction

There is an urgent need to study various aspects of pathogenic fungi since the latest estimates on invasive fungal infections report an annual infection incidence of 6.5 million and 3.8 million deaths of which 2.8 million are directly attributed to fungal infections [1]. Pathogenic yeasts are especially concerning, and the recently published fungal priority pathogens list [2] includes several pathogenic yeasts from the genera *Candida* and *Cryptococcus*. In addition, antifungal resistance has become an increasing global concern [3] and makes the effective treatment of fungal infections difficult [4]. Numerous studies have reported that drug resistance of certain clinical fungal isolates was of environmental origin [5,6].

Microorganisms and chemicals can enter water systems by being released by infected individuals, washed from household surfaces or through agricultural or industrial run-off. Wastewater treatment plants (WWTPs) collect microorganisms, pharmaceuticals, and heavy metals [7,8,9,10] and may provide valuable information regarding the health and lifestyle of a community in a specific catchment area which can be accessed via wastewater-surveillance [11,12].

Wastewater surveillance entails monitoring wastewater for specific agents of interest (e.g., bacteria and viruses) to determine the overall health of a community. The potential of wastewater surveillance has long been known and applied to monitoring polio outbreaks [13,14,15] and in recent times to monitor COVID-19 infection levels [11,16,17,18], yet limited attempts have been made to apply this system to the surveillance of pathogenic yeast [19,20,21,22].

This study aims to address this gap, by combining culture-dependent and a culture independent multiplex PCR system to evaluate the potential application of yeast wastewater surveillance. Additionally, the ability of WWTPs to support yeast proliferation will be investigated.

## 2. Materials and Methods

### 2.1. Sample Collection

Sampling was done once a month from May 2023 to April 2024. Grab samples were collected from six different sites in Bloemfontein, Mangaung Metro, South Africa (Appendix A) according to the protocol outlined by the South African Medical Research Council [23]. One liter of wastewater influent was collected after the first grid screen, stored in closed, labelled containers during transport and kept at 4 °C until use (within 2 h after the last sample was collected). Information regarding the population group and geographical area covered by each sample site can be found in Appendix A.

Samples could not be collected from University of the Free State (UFS) from January 2024 to April 2024 due to construction blocking the manhole used to access raw wastewater. Sterkwater (SW) treatment plant was not operational during June 2023 and October 2023 for various reasons, and no samples were collected from Bainsvlei treatment plant (BV) for November 2023 due to power failures.

### 2.2. Yeast Isolation and Identification

Yeast isolation was performed every second month from May 2023 to April 2024. A 50 mL aliquot of wastewater was centrifuged at 6854× *g* (Centrifuge 5430R Eppendorf; Johannesburg, South Africa: F-35-6-30 rotor) and the supernatant was diluted 100 times. These dilutions were filtered using a 0.45 µm cellulose nitrate filter (Satorius, Göttingen, Germany) and the filter disks placed on Sabouraud dextrose agar, supplemented with chloramphenicol (10 g/L peptone powder, 40 g/L dextrose, 500 mg/L chloramphenicol, 15 g/L bacteriological agar—Merck, Modderfontein, South Africa) and incubated at 37 °C for 48 h. Six yeast colonies were selected using Harrison’s Disc method [24] and further purified on yeast malt agar (YM) (3 g/L yeast extract, 3 g/L malt extract, 5 g/L peptone powder, 10 g/L glucose and 16 g/L bacteriological agar) and incubated at 37 °C. Colonies were confirmed to be yeast cells using microscopy and the isolates were deposited into the SANBI Biodiversity Biobank SA yeast culture collection at the Department of Microbiology and Biochemistry, University of the Free State, Bloemfontein, South Africa.

For all pure cultures, colony PCR targeting the ITS region was performed. Briefly, a small amount of an overnight grown colony was resuspended into 10.5 µL of molecular-grade water and boiled at 94 °C for 10 min in a thermocycler (Applied Biosystems 2720 Thermocycler; Applied Biosystems, Randburg, South Africa). PCR was performed using ITS 4 primer (-5′-TCCTCCGCTTATT3ATATGC-3′), ITS 5 primer (5′-GGAAGTAAAAGTCGTAACAAGG-3′) (Integrated DNA Technologies, Coralville, IA, USA) and Quantabio AccuStart II PCR ToughMix^®^ [2X] (WhiteSci, Cape Town, South Africa). The PCR conditions were as follows: Initial denaturation at 94 °C for 5 min, denaturation at 94 °C for 30 s, annealing at 50 °C for 30 s, extension at 72 °C for 30 s, Final extension at 72 °C for 5 min and hold at 4 °C. Amplification of the ITS sequence was confirmed by gel electrophoresis.

Amplicons were cleaned with New England Biolabs FastAP™ Thermosensitive Alkaline Phosphatase (Thermo Fischer Scientific, Johannesburg, South Africa) and New England Biolabs Exonuclease I (Thermo Fischer Scientific, Johannesburg, South Africa) and incubated at 37 °C for 15 min and a further 15 min at 85 °C. Clean-up products were prepared for sequencing using the Applied Biosystems BigDye^TM^ Terminator v. 3.1 Cycle Sequencing Kit (Thermo Fischer Scientific, Johannesburg, South Africa) The PCR conditions were as follows: Initial denaturation at 96 °C for 1 min, denaturation at 96 °C for 10 s, annealing at 50 °C for 5 s, extension at 60 °C for 4 min and a hold phase at 4 °C, the reaction was set for 25 cycles.

For post-reaction clean-up, the 10 µL reaction volume was adjusted to 20 µL using molecular grade water. This was then transferred to 1.5 mL Eppendorf tubes containing 5 µL [125 mM] EDTA and 60 µL absolute ethanol. The tubes were vortexed and left to precipitate at room temperature for 15 min, and centrifuged at 20,000× *g* for 15 min at 4 °C. The supernatant was completely aspirated without disturbing the pellet and 200 µL of 70% ethanol was added, the tubes were centrifuged at 20,000× *g* for 5 min at 4 °C. The supernatant was then completely aspirated, and the pellet was dried in an Eppendorf Speed-Vac (Eppendorf Concentrator Plus) (Johannesburg, South Africa) for 5 min.

Amplicons were sequenced using Applied Biosystems™ 3500 genetic analyzer (Thermo Fischer, Johannesburg, South Africa). Geneious Prime version 2023.2.1 was used to construct a consensus sequence. This consensus sequence was identified using NCBI’s BLAST function.

### 2.3. Growth Studies

One representative isolate from each of the identified species was cultured overnight on YM agar at 37 °C for 24 h. Isolates were standardized with sterile water to 1 × 10^−6^ cells/mL. Wastewater from the Bloemwater (BW) treatment plant was filtered using glass fibre (Schleicher and Schuell Bioscience, Keene, NH, USA), 0.45 µm nitrocellulose (Satorius, Göttingen, Germany) and 0.22 µm filter discs and 90 µL dispensed into a 96-well plate followed by 10 µL of standardised inoculum (final concentration of 1 × 10^−5^ cells/mL). The Victor Nivo Multimode Microplate Reader (Revvity, Biocom Africa, Pretoria, South Africa) was used to measure optical density (OD_600nm_) every hour for 48 h at 37 °C with orbital shaking before every measurement. This was done in biological and technical triplicates and the averages plotted. Uninoculated filtered wastewater served as negative control.

### 2.4. DNA Extraction from Wastewater

A 5 mg/mL stock of photoactivated ethidium monoazide bromide (Thermo Fischer Scientific, Johannesburg, South Africa), dissolved in a 20% dimethyl sulfoxide (Merck, Modderfontein, South Africa), was added to a 50 mL wastewater sample to a final concentration of 6 µM [25]. Samples were vortexed and incubated horizontally on ice for 10 min in the absence of light. Incubation was followed by a 15-min exposure to a halogen light source. Samples were washed with 1 mL 0.85% NaCl solution and centrifuged at 7830 rpm (Centrifuge 5430R Eppendorf^®^ USA: F-35-6-30 rotor) after which the supernatant was discarded, and the remaining pellet was resuspended in 30 mL PBS. DNA from living cells was extracted using the Omega Bio-Tek E.N.Z.A.^®^ Water DNA kit (WhiteSci, Cape Town, South Africa). Wastewater (50 mL) was filtered through a 0.45 µm cellulose nitrate filter and DNA extraction performed according to the manufacturer’s instructions with elution using 60 µL of elution buffer. Gel electrophoresis was used to confirm genomic DNA extraction.

### 2.5. Multiplex PCR

This was performed according to the protocol by Arastehfar and colleagues [26] with some modifications. Four reactions were set up. Multiplex 1 contained the primers for *Candida albicans, Candida auris*, *Candida dubliniensis*, *Candida tropicalis*, *Candida glabrata*, *Candida parapsilosis* and *Candida krusei* (*Pichia kudriavzevii*). Multiplex 2a contained primers for *Myerozyma guiliermondii*, *Kluyveromyces marxianus* and *Clavispora lusitaniae.* Multiplex 2b contained primers for *Debaromyces hansenii*, *Yarrowia lipolytica*, *Pichia norvegensis* and *Diutina rugosa*. Multiplex 3 contained primers for *Cryptococcus neoformans*, *Cryptococcus deneoformans*, *Cryptococcus gattii*, *Rhodotorula mucilaginosa*, *Geotrichum candidum*, *Trichosporon asahii* and *Trichosporon lactis.* Each reaction contained 5 µL DNA template, 1 µL forward and reverse primer, 1 µL of 10 mM dNTPs, 0.5 µL New England Biolabs Q5^®^ High-Fidelity DNA Polymerase ( Inqaba Biotechnical Industries, Pretoria, South Africa), 10 µL of a (5×) reaction buffer and molecular grade water up to 50 µL final volume. PCR conditions were as follows: initial denaturation at 95 °C for 5 min (1 cycle), denaturation at 95 °C for 30 s, annealing at 50 °C for 30 s and extension at 72 °C for 40 s (each for 35 cycles), and lastly final extension at 72 °C for 8 min. PCR amplicons were visualized using gel electrophoresis and identified based on amplicon size as specified by Arastehfar and colleagues [26] (Appendix A).

## 3. Results

### 3.1. Culture-Dependant Surveillance Indicate the Presence of Pathogenic Yeast Species

One hundred-and-sixty-six (166) isolates were obtained and identified. It was found that *Candida* species made up 43.98% of the total isolates (Table 1). This corresponds to previous studies investigating fungal occurrence in various freshwater sources as well as tap water [27,28,29,30,31]. In these studies, the most abundant *Candida* species were *C. tropicalis* [28,32] and *C. parapsilosis* [27,31]. Studies investigating fungal presence in wastewater tend to focus on species occurring in activated sludge and found *C. albicans* and *C. krusei* [33,34,35,36]. In this study *C. krusei* was isolated the most often followed by *C. glabrata*. The frequent occurrence of *C. glabrata* was expected as it is frequently found in the human gastrointestinal tract and nature but the prominent presence of *C. krusei* was surprising.

### 3.2. Several Species Can Grow in Wastewater

To determine if the ability of the isolated yeast species to grow in the wastewater during transit through the system can contribute to their cell number and likelihood of isolation, the ability of a representative isolate from each species was tested for growth in a pooled wastewater sample. Yeast with similar growth curves were grouped together (Figure 1). For some species there was agreement between the frequency of isolation of and their ability to grow in wastewater. *C. krusei*, the most often isolated species, grew the best, reaching an OD_600_ of 0.38 within 24 h (Figure 1A). Similarly, *C. glabrata*, the second most isolated species, grew to a final OD_600_ of 0.27. This could point to their abundant occurrence possibly being linked to their ability to grow well in wastewater. In contrast, *Magnusiomyces clavatus* (previously known as *Geotrichum clavatum* and *Saprochaete clavata*) and *Candida palmioleophila* were not often isolated, although they were also able to grow in wastewater (Figure 1A). Although the ecology of *M. clavatus* is not well studied, other *Magnusiomyces* spp. are ubiquitous in the environment and the digestive tract of animals and humans [37]. *C. palmioleophila* is an environmental yeast with high lipolytic activity, originally isolated from palm oil [38]. Interestingly, this opportunistic pathogen [39] was previously isolated from hospital wastewater [40].

*Saccharomyces cerevisiae*, *Hanseniaspora pseudoguilliermondii* and *Sporopachydermina lavtivora* were not able to grow in the wastewater and both *H. pseudoguilliermondii* and *S. lavtivora* were only isolated once, indicating that wastewater may not be an ideal habitat for these yeasts. Interestingly, *S. cerevisiae* was the third most isolated species. The fact that it seems unable to grow in the wastewater, suggests multiple independent introductions of this yeast to the wastewater system over the 12-month period.

### 3.3. Culture-Independent Surveillance Identifies Various Pathogenic Yeasts

Figure 2 depicts the prevalence of the detection of the different targeted species. Two species, *C. dubliniensis* and *T. lactis* were not detected, while *C. glabrata* was the most prevalent, occurring in 98.5% of all samples, followed by *C. lusitaniae* (96.9%) and *C. tropicalis* (92.3%). In addition, among the *Cryptococcus* spp., *Cryptococcus neoformans* and *Cryptococcus deneoformans* were detected in more samples than *Cryptococcus gattii*. Bloemfontein falls in a semi-arid climate with hot, wet summers and cold, dry winters. Thus, the dry months are June to August and the wet months are December to February. However, no seasonal variation or specific patterns of occurrence in the different WWTPs were observed (Appendix A).

## 4. Discussion

More than 90% of candidemia cases are attributed to only five *Candida* species, namely *C. albicans*, *C. glabrata*, *C. krusei*, *C. tropicalis* and *C. parapsilosis* [41] with *C. albicans* being the predominant causative agent. In the present work, the most frequently isolated species was *C. krusei*, an emerging nosocomial fungal pathogen [42,43] which is grouped in the medium-risk group of the recently released fungal priority pathogens list [18]. Among the top five pathogenic *Candida* species, *C. krusei* is the least well-studied and normally uncommon in the human gut microbiome [44] but widespread in nature. Importantly, it is intrinsically resistant to fluconazole [45]. There also seems to be a strong genetic relatedness between environmental strains and clinical strains [44], which begs the question if humans are acquiring *C. krusei* infections from the environment or if clinical strains are polluting the environment.

It may also be that *C. krusei* is particularly able to grow within this environment since species that were most frequently isolated (*C. krusei* and *C. glabrata*) grew well in wastewater, while others that were isolated once showed poor to no growth. However, other factors may play a role in determining which yeasts are present in wastewater rather than just their ability to grow in this environment.

The ability of wastewater to support yeast proliferation becomes a concern since the Free State province (which includes Mangaung Metropole) has extremely poor maintenance of WWTPs and was ranked with the third worst overall report according to the 2019 Green Drop report for WWTPs in South Africa. This ranking was given due to 44% of WWTP in this province being in a critical state and Mangaung was one of the top three contributing municipalities. Thus, WWTPs are often non-functional leading to plants being flooded and water remaining stagnant for long periods. This stagnant water may contribute to yeast proliferation and the spread of potentially harmful yeast species to the surrounding environment.

Most attempts that have been made to detect fungal pathogens through culture-independent techniques in wastewater, targeted a single yeast species, namely *C. auris* [19,20,21,22] even though various tools, including multiplex PCR, are available to make the detection of multiple target species in one sample a possibility [26,46]. We presented the potential application of a wastewater surveillance system for multiple pathogenic yeast species instead of just a single target. It was determined that a wide range of potentially pathogenic yeast species can be found in the influent of WWTPs with certain *Candida* species being detected by both culture-dependent and independent methods. A recent study, screening wastewater for SARS-CoV-2 and yeasts, using polyphasic taxonomy, identified *C. albicans*, *C. krusei*, *C. palmioleophila*, *C. tropicalis* and *C. utilis,* to be present [47] (Corrêa-Moreira et al., 2024). Except for *C. utilis,* these species were also detected in our study using both methods. Further agreement between the culture-dependent and -independent data can be seen by the identification of *C. glabrata* as a prominent species in wastewater by both methods. This corresponds with the study of Steffen and colleagues [48] who found that *C. glabrata* is associated with contamination of water.

Interestingly, *C. krusei*, which was the most isolated species, was only detected using multiplex PCR in 15.4% of the samples. A possible explanation for this discrepancy is that the specific primers used to detect this species may not be sensitive enough, although it was previously found not to be a problem in clinical samples [26]. In addition, no basidiomycetous yeasts were isolated, although they were detected using the multiplex PCR, and we were also unable to isolate *C. auris*, although this yeast was detected by multiplex PCR in 36.4% of samples. This may indicate that the culture-independent technique is more sensitive to species that may occur in low cell numbers in the samples, especially since the Harrison’s Disc method only selects six yeast colonies and is thus statistically more likely to isolate colonies belonging to more abundant species or species better able to grow under laboratory conditions used.

From previous studies investigating the application of wastewater surveillance to pathogenic yeast, the correlation between *C. auris* infections and their levels in wastewater is still unclear [19,20,21,22]. Thus, more research comparing the occurrence and levels of pathogenic yeast in wastewater to their occurrence in clinical settings is required to validate wastewater surveillance as an efficient and accurate community monitoring system for pathogenic yeast outbreaks. A complication of this surveillance system is that some of the yeast present in wastewater naturally occur in the environment and are not necessarily shed from infected individuals, thus care should be taken when interpreting the data collected from these studies. In addition, certain yeasts may form more permanent biofilms within the system [49], which may act as a constant source of these yeasts. This would complicate any time-based correlation between the presence of the yeast in the community and hospital infections. However, the presence of *C. glabrata* in wastewater may be used as a potential indicator of fecal contamination of water and should be investigated further.

## 5. Conclusions

WWTPs may act as a hub for pathogenic yeast strains with *Candida* species the most frequently isolated and identified using culture-independent and dependent techniques. A multiplex PCR system could aid in the timely detection of various pathogens circulating within the community of a specific catchment area. Growth studies reveal the potential role WWTPs may play in supporting yeast proliferation and spread. Regardless of how the strains arrive in the wastewater, they may pose serious health risks to susceptible persons that encounter this water, as part of their work or due to poor infrastructure leading to sewage spills. Overall wastewater surveillance of pathogenic yeast shows promise and should be explored further.

## Figures and Tables

**Figure 1 jof-11-00086-f001:**
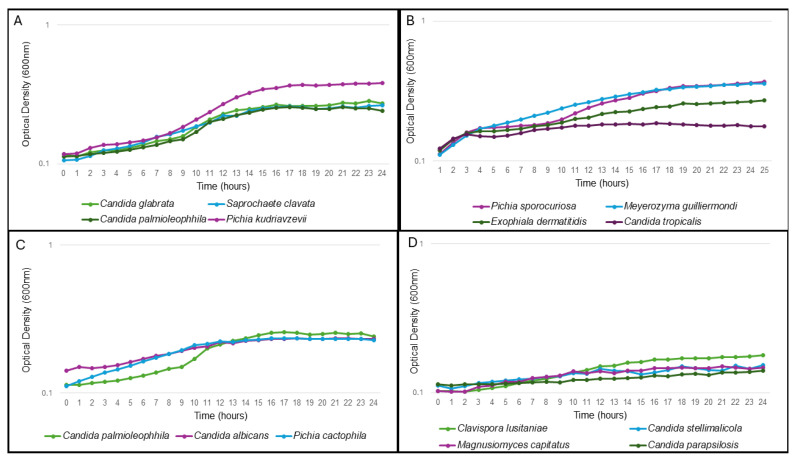
Growth of representative strains of each isolated species in filter sterilized wastewater. Species with similar growth characteristics are grouped together. Values represent the average of three independent experiments. In all cases, the standard deviations were ≤10% of the average. (**A**) contains the growth curves of *Candida glabrata*, *Candida palmioleophila*, *Pichia kudriavzevii* and *Saprochaete clavate*. (**B**): *Pichia sporocuriosa*, *Meyerozyma guilliermondi*, *Exophiala dermatitidis* and *C. tropicalis*. (**C**): *Candida palmioleophila*, *Candida albicans* and *Pichia cactophila*. (**D**): *Clavispora lusitaniae*, *Candida parapsilosis*, *Magnusiomyces capitatus* and *Candida stellimalicola*.

**Figure 2 jof-11-00086-f002:**
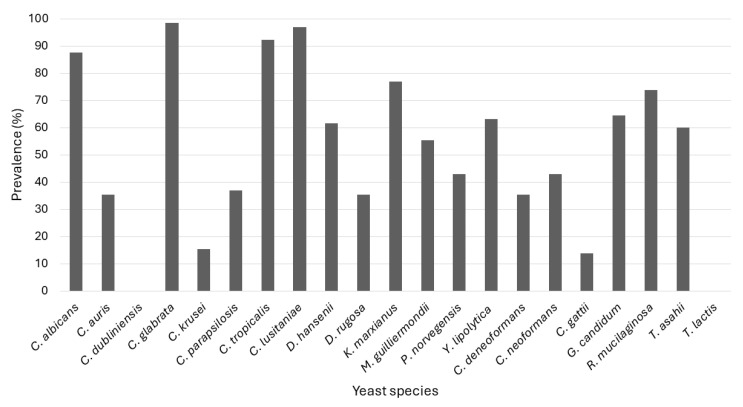
Yeast prevalence over the 12-month sampling period, according to the data from the multiplex PCRs.

**Table 1 jof-11-00086-t001:** Yeast species and number of isolates obtained from May 2023 to March 2024.

Species	05/2023	07/2023	09/2023	11/2023	01/2024	03/2024	Total No.
*Candida albicans*	4	3	6	0	2	3	18
*Candida glabrata*	11	6	8	3	3	10	41
*Candida palmioleophila*	1	1	0	0	0	0	2
*Candida parapsilosis*	0	0	0	0	1	0	1
*Candida stellimalicola*	0	1	0	0	0	0	1
*Candida tropicalis*	0	3	1	0	4	0	8
*Candida* sp.	0	0	2	0	0	0	2
*Clavispora lusitaniae*	5	0	0	0	1	0	6
*Dipodascus capitatus*	0	0	0	1	0	0	1
*Exophiala dermatitidis*	0	0	0	1	0	0	1
*Hanseniaspora pseudoguilliermondii*	0	0	1	0	0	0	1
*Magnusiomyces capitatus*	0	0	0	1	0	1	2
*Meyerozyma guilliermondii*	1	0	0	0	0	0	1
*Pichia cactophila*	0	1	0	0	0	0	1
*Pichia kudriavzevii*/*C. krusei*	2	4	5	13	11	10	45
*Pichia sporocuriosa*	0	0	0	0	1	0	1
*Saccharomyces cerevisiae*	4	9	10	5	3	0	31
*Magnusiomyces clavatus*	0	0	0	1	0	0	1
*Sporopachydermia lactativora*	0	1	0	1	0	0	2

## Data Availability

All data is available within this manuscript and Appendix A.

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
