# Peer review of "Culture-Dependent and -Independent Wastewater Surveillance for Multiple Pathogenic Yeasts"

_jof, 2025, doi:10.3390/jof11020086_

Round 1

Reviewer 1 Report

Data on wastewater surveillance are scarce as such the results presented by Tyla Baker are interesting for the scientific public. There are however several methodological issues, as such I propose to remove several parts of the manuscript and to entirely restructure the results and discussion section.

Major comments

1.       The abstract does not clearly summarize the most important results obtained and should include more concrete data.

2.       The introduction should end with clearly stating the aims of the study.

3.       Fluconazole susceptibility testing. According to the authors fluconazole susceptibility testing was performed according to an outdated EUCAST protocol (from 2008!) en not from an freely online available more recent version. You can only state that susceptibility testing was performed according to EUCAST if the procedure was exactly followed as in the protocol which does not seem to be the case (inoculum preparation, wavelength for reading etc.). The content of Table 6 is not clear. It looks like MIC50, MIC90 en MIC were calculated for individual isolates whereas MIC50 and MIC90 are values calculated for populations tested and not for individual isolates. In addition, the number of isolates tested is too low to draw any conclusions. Because of all these issues I propose to remove the fluconazole susceptibility part testing from the manuscript.

4.       The authors tried to correlate the wastewater results with diagnosed yeast infections. This part however suffers from several issues. Line 200, to calculated fungal infection incidences denominator data are needed which are not available. There are no case definitions and the majority of the Candida species cultured from patients were not specified. As such I propose to remove the correlation with the clinical data from the manuscript.

5.       The majority of the text of the results section does not below to this section but to the discussion section (comparison with literature and interpretation of the data), as such the manuscript should be restructured.

6.       Figure 4. It is unclear how total fluconazole concentrations were calculated. It seems that concentrations were added up which is methodologically not correct. Instead the maximal detected concentrations could be presented.

7.       Lines 392-395. The authors conclude that sensitivity of the multiplex PCR is higher than that of the culture-based detection method. Please reflect on the impact of the methodology used (selection of 6 isolates by the Harrison’s Disc method) on the sensitivity of the culture-based method.

8.       Please reflect more in depth (in the discussion section) on the different results obtained for C. glabrata and C. krusei comparing culture and PCR.

Minor comments

1.       Materials and methods:

a.       Line 62. Write SW in full (first time stated in the text)

b.       Line 67. State the centrifuge rate in g-value (can be calculated)

c.       Mention clearly that identification of the yeasts was conducted by ITS sequencing

2.       The sentence on lines 328-330 is unclear for me. What is meant with ‘this lower trend’ and why does this indicate that WWTP could act as concentrators of fluconazole?

Reviewer 2 Report

This paper is very interesting and deserves publication after revision the Materials and Methods section should be shortened and focus on the main points. The authors should respond to the following comments:

Comments

1.        Table 1 - should be shortened or moved to supplementary.

2.        Line 67 - give the x g .

3.        Line 138 - write the full name of the species – Candida glabrata.

4.        Table 2  - should be part of the supplementary.

5.        Table 3 -  is too short for a table. It can be described in words.

6.        The section of materials and methods should be shortened and focus on the main points.

7.        Figure 1 -  write down in the legends what each panel represents?

8.        Line 274  0 C. neoformans and C. deneoformans give the full name of the species.

9.        Table 5 -  It is suggested that Table 5 be removed and replaced with textual content.

10.   Line 392 - It should be noted in the discussion that the multiplex PCR does not distinguish between dead yeast and live yeast

Round 2

Reviewer 1 Report

De authors responded well to the comments raised.

I have no further comments